# The nature of the animacy organization in human ventral temporal cortex

Sushrut Thorat[1]*, Daria Proklova[2], Marius V Peelen[1]*

[1]Donders Institute for Brain, Cognition and Behaviour, Radboud University, Nijmegen, Netherlands; [2]Brain and Mind Institute, University of Western Ontario, London, Canada

**Abstract** The principles underlying the animacy organization of the ventral temporal cortex (VTC) remain hotly debated, with recent evidence pointing to an animacy continuum rather than a dichotomy. What drives this continuum? According to the visual categorization hypothesis, the continuum reflects the degree to which animals contain animal-diagnostic features. By contrast, the agency hypothesis posits that the continuum reflects the degree to which animals are perceived as (social) agents. Here, we tested both hypotheses with a stimulus set in which visual categorizability and agency were dissociated based on representations in convolutional neural networks and behavioral experiments. Using fMRI, we found that visual categorizability and agency explained independent components of the animacy continuum in VTC. Modeled together, they fully explained the animacy continuum. Finally, clusters explained by visual categorizability were localized posterior to clusters explained by agency. These results show that multiple organizing principles, including agency, underlie the animacy continuum in VTC.

DOI: https://doi.org/10.7554/eLife.47142.001

## Introduction

One of the main goals of visual cognitive neuroscience is to understand the principles that govern the organization of object representations in high-level visual cortex. There is broad consensus that the first principle of organization in ventral temporal cortex (VTC) reflects the distinction between animate and inanimate objects. These categories form distinct representational clusters (*Kriegeskorte et al., 2008*) and activate anatomically distinct regions of VTC (*Grill-Spector and Weiner, 2014*; *Chao et al., 1999*).

According to the visual categorization hypothesis, this animate-inanimate organization supports the efficient readout of superordinate category information, allowing for the rapid visual categorization of objects as being animate or inanimate (*Grill-Spector and Weiner, 2014*). The ability to rapidly detect animals may have constituted an evolutionary advantage (*Caramazza and Shelton, 1998*; *New et al., 2007*).

However, recent work has shown that the animacy organization reflects a continuum rather than a dichotomy, with VTC showing a gradation from objects and insects to birds and mammals (*Connolly et al., 2012*; *Sha et al., 2015*). This continuum was interpreted as evidence that VTC reflects the psychological property of animacy, or agency, in line with earlier work showing animate-like VTC responses to simple shapes whose movements imply agency (*Castelli et al., 2002*; *Martin and Weisberg, 2003*; *Gobbini et al., 2007*). According to this agency hypothesis, the animacy organization reflects the degree to which animals share psychological characteristics with humans, such as the ability to perform goal-directed actions and experiencing thoughts and feelings.

Importantly, however, the finding of an animacy continuum can also be explained under the visual categorization hypothesis. This is because some animals (such as cats) are easier to visually categorize as animate than others (such as stingrays). This visual categorizability is closely related to visual

*For correspondence:
s.thorat@donders.ru.nl (ST);
m.peelen@donders.ru.nl (MVP)

**Competing interests:** The authors declare that no competing interests exist.

typicality – an animal's perceptual similarity to other animals (*Mohan and Arun, 2012*). Indeed, recent work showed that the visual categorizability of animals (as measured by reaction times) correlates with the representational distance of those animals from the decision boundary of an animate-inanimate classifier trained on VTC activity patterns (*Carlson et al., 2014*). The finding of an animacy continuum is thus fully in line with the visual categorization hypothesis.

The difficulty in distinguishing between the visual categorization and agency hypotheses lies in the fact that animals' visual categorizability and agency are correlated. For example, four-legged mammals are relatively fast to categorize as animate and are also judged to be psychologically relatively similar to humans. Nevertheless, visual categorizability and agency are distinct properties that can be experimentally dissociated. For example, a dolphin and a trout differ strongly in perceived agency (dolphin > trout) but not necessarily in visual categorizability. In the present fMRI study, we disentangled visual categorizability and agency to assess their ability to explain the animacy continuum in VTC. This was achieved by selecting, out of a larger set, 12 animals for which visual categorizability and agency were orthogonal to each other across the set.

We find that visual categorizability and agency independently contribute to the animacy continuum in VTC as a whole. A model that combines these two factors fully explains the animacy continuum. In further analyses, we localize the independent contributions of visual categorizability and agency to distinct regions in posterior and anterior VTC, respectively. These results provide evidence that multiple organizing principles, including agency, underlie the animacy continuum and that these principles express in different parts of visual cortex.

## Results

### Disentangling visual categorizability and agency

To create a stimulus set in which visual categorizability and agency are dissociated, we selected 12 animals from a total of 40 animals. Visual categorizability was quantified in two ways, using convolutional neural networks (CNNs) and human behavior, to ensure a comprehensive measure of visual categorizability. Agency was measured using a rating experiment in which participants indicated the degree to which an animal can think and feel. Familiarity with the objects was also assessed and controlled for in the final stimulus set used in the fMRI experiment.

#### Agency and familiarity

Agency and familiarity measures were obtained through ratings (N = 16), in which participants indicated the thoughtfulness of, feelings of, and familiarity with the 40 animals (*Figure 1A*). The correlation between the thoughtfulness and feelings ratings ($\tau = 0.70$) was at the noise ceiling of both those ratings ($\tau_{thought} = 0.69$, $\tau_{feel} = 0.70$). We therefore averaged the thoughtfulness and feelings ratings and considered the averaged rating a measure of agency.

#### Visual categorizability

The first measure of visual categorizability was based on the features extracted from the final layer (FC8) of a pre-trained CNN (VGG-16 [*Simonyan and Zisserman, 2015*]; Materials and methods). This layer contains rich feature sets that can be used to accurately categorize objects as animate or inanimate by a support vector machine (SVM) classifier. This same classifier was then deployed on the 40 candidate objects (4 exemplars each) of our experiment to quantify their categorizability. This resulted, for each object, in a representational distance from the decision boundary of the animate-inanimate classifier (*Figure 1B*). Because this measure was based on a feedforward transformation of the images, which was not informed by inferred agentic properties of the objects (such as thoughtfulness), we label this measure image categorizability.

The second measure of visual categorizability was based on reaction times in an oddball detection task previously shown to predict visual categorization times (*Mohan and Arun, 2012*; *Figure 1C*). The appeal of this task is that it provides reliable estimates of visual categorizability using simple and unambiguous instructions (unlike a direct categorization task, which relies on the participants' concept of animacy, again potentially confounding agency and visual categorizability). Participants were instructed to detect whether an oddball image appears to the left or the right of fixation. Reaction times in this task are an index of visual similarity, with relatively slow responses to

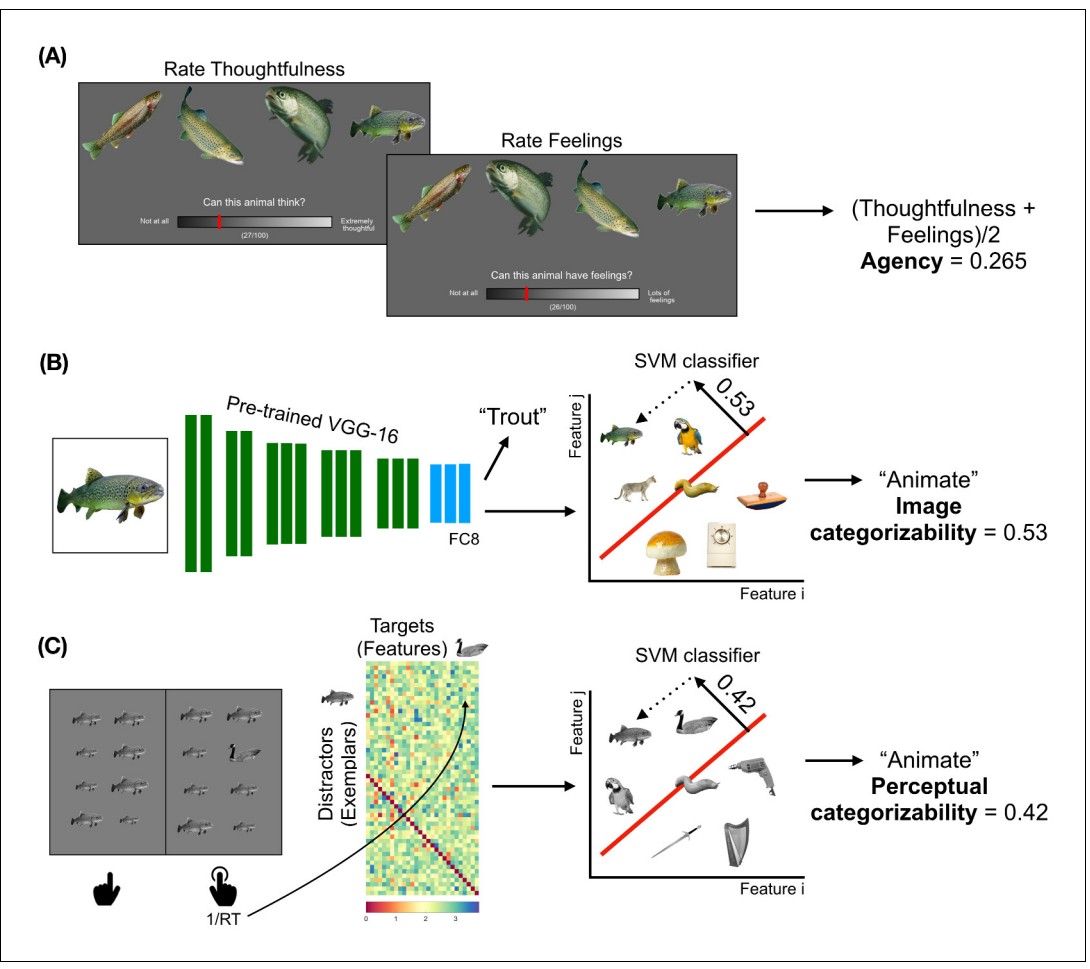

**Figure 1.** Obtaining the models to describe animacy in the ventral temporal cortex. (**A**) Trials from the ratings experiment are shown. Participants were asked to rate 40 animals on three factors - familiarity, thoughtfulness, and feelings. The correlations between the thoughtfulness and feelings ratings are at the noise ceilings of both these ratings. Therefore, the average of these ratings was taken as a measure of agency. (**B**) A schematic of the convolutional neural network (CNN) VGG-16 is shown. The CNN contains 13 convolutional layers (shown in green), which are constrained to perform the spatially-local computations across the visual field, and three fully-connected layers (shown in blue). The network is trained to take RGB image pixels as inputs and output the label of the object in the image. Linear classifiers are trained on layer FC8 of the CNN to classify between the activation patterns in response to animate and inanimate images. The distance from the decision boundary, toward the animate direction, is the image categorizability of an object. (**C**) A trial from the visual search task is shown. Participants had to quickly indicate the location (in the left or right panel) of the oddball target among 15 identical distractors which varied in size. The inverse of the pairwise reaction times are arranged as shown. Either the distractors or the targets are assigned as features of a representational space on which a linear classifier is trained to distinguish between animate and inanimate exemplars (Materials and methods). These classifiers are then used to categorize the set of images relevant to subsequent analyses; the distance from the decision boundary, towards the animate direction, is a measure of the perceptual categorizability of an object.
DOI: https://doi.org/10.7554/eLife.47142.002

The following figure supplement is available for figure 1:

**Figure supplement 1.** Pairwise similarities between the image categorizabilities of layers from VGG-16.
DOI: https://doi.org/10.7554/eLife.47142.003

oddball objects that are visually relatively similar to the surrounding objects (e.g. a dog surrounded by cats). A full matrix of pairwise visual similarities was created by pairing all images with each other. For a given object, these similarity values constitute a perceptual representation with respect to the other objects. These visual similarity values were then used as features in an SVM trained to classify

animate vs inanimate objects. Testing this classifier on the images of the fMRI experiment resulted, for each object, in a representational distance from the decision boundary of the animate-inanimate classifier (*Figure 1C*). Because this measure was based on human perception, we labeled this measure perceptual categorizability. The neural representations the reaction times in this task rely on are not fully known, and might reflect information about inferred agency of the objects. As such, accounting for the contribution of perceptual categorizability in subsequent analyses provides a conservative estimate of the independent contribution of agency to neural representations in VTC.

The two measures of visual categorizability were positively correlated for the 12 animals that were selected for the fMRI experiment (Kendall's $\tau = 0.64$), indicating that they partly reflect similar animate-selective visual properties of the objects. The correspondence between these two independently obtained measures of visual categorizability provides a validation of these measures and also shows that the image categorizability obtained from the CNN is meaningfully related to human perception.

## Selection of image set

The final set of 12 animals for the fMRI experiment were chosen from the set of 40 images such that the correlations between image categorizability, agency, and familiarity were minimized (*Figure 2*). This was successful, as indicated by low correlations between these variables ($\tau < 0.13$, for all correlations). Because perceptual categorizability was not part of the selection procedure of the stimulus set, there was a moderate residual correlation ($\tau = 0.30$) between perceptual categorizability and agency.

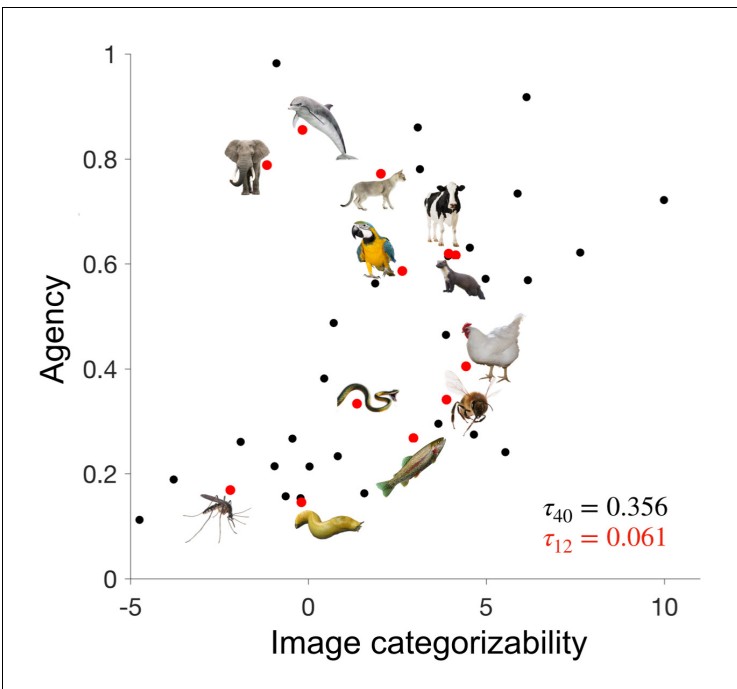

**Figure 2.** Disentangling image categorizability and agency. The values of agency and image categorizability are plotted for the 40 animals used in the ratings experiment. We selected 12 animals such that the correlation between agency and image categorizability is minimized. Data-points corresponding to those 12 animals are highlighted in red.
DOI: https://doi.org/10.7554/eLife.47142.004

The following source data is available for figure 2:

**Source data 1.** Agency and image categorizability scores for the 40 animals.
DOI: https://doi.org/10.7554/eLife.47142.005

## Animacy in the ventral temporal cortex

Participants (N = 17) in the main fMRI experiment viewed the 4 exemplars of the 12 selected animals while engaged in a one-back object-level repetition-detection task (*Figure 3*). The experiment additionally included 3 inanimate objects (cars, chairs, plants) and humans. In a separate block-design animacy localizer experiment, participants viewed 72 object images (36 animate, 36 inanimate) while detecting image repetitions (*Figure 3*).

In a first analysis, we aimed to replicate the animacy continuum for the objects in the main experiment. The VTC region of interest was defined anatomically, following earlier work (*Haxby et al., 2011*; *Figure 4A*; Materials and methods). An SVM classifier was trained on activity patterns in this region to distinguish between blocks of animate and inanimate objects in the animacy localizer, and tested on the 16 individual objects in the main experiment. The distances from the decision boundary, towards the animate direction, were taken as the animacy scores.

The mean cross-validated training accuracy (animacy localizer) of animate-inanimate classification in VTC was 89.6%, while the cross-experiment accuracy in classifying the 16 stimuli from the main fMRI experiment was 71.3%, indicating reliable animacy information in both experiments. Importantly, there was systematic and meaningful variation in the animacy scores for the objects in the main experiment (*Figure 4B*). Among the animals, humans were the most animate whereas reptiles and insects were the least animate (they were classified as inanimate, on average). These results replicate previous findings of an animacy continuum (*Connolly et al., 2012*; *Sha et al., 2015*).

Now that we established the animacy continuum for the selected stimulus set, we can turn to our main question of interest: what are the contributions of visual categorizability and agency to the animacy continuum in VTC? To address this question, we first correlated the visual categorizability scores and the agency ratings with the VTC animacy scores (*Figure 4C*). VTC animacy scores correlated positively with all three measures: image categorizability ($\tau = 0.16$; $p = 10^{-3}$), perceptual categorizability ($\tau = 0.26$; $p < 10^{-4}$); and agency ($\tau = 0.30$; $p < 10^{-4}$). A combined model of image categorizability and perceptual categorizability (*visual categorizability*; Materials and methods) also positively correlated with VTC animacy ($\tau = 0.23$; $p < 10^{-4}$).

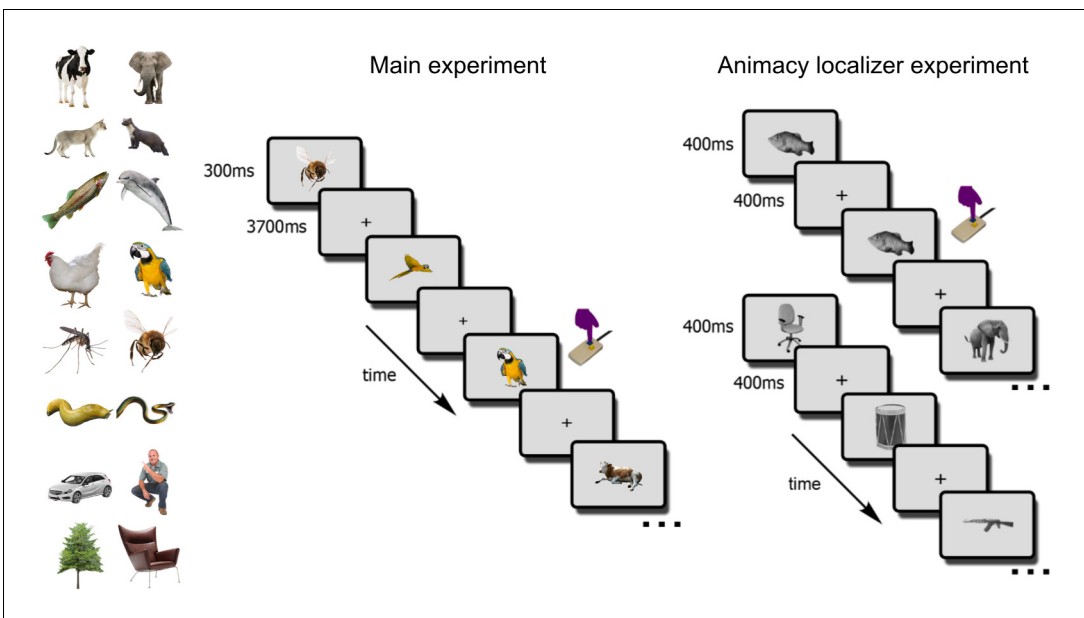

**Figure 3.** The fMRI paradigm. In the main fMRI experiment, participants viewed images of the 12 selected animals and four additional objects (cars, trees, chairs, persons). Participants indicated, via button-press, one-back object repetitions (here, two parrots). In the animacy localizer experiment, participants viewed blocks of animal (top sequence) and non-animal (bottom sequence) images. All images were different from the ones used in the main experiment. Each block lasted 16s, and participants indicated one-back image repetitions (here, the fish image).
DOI: https://doi.org/10.7554/eLife.47142.006

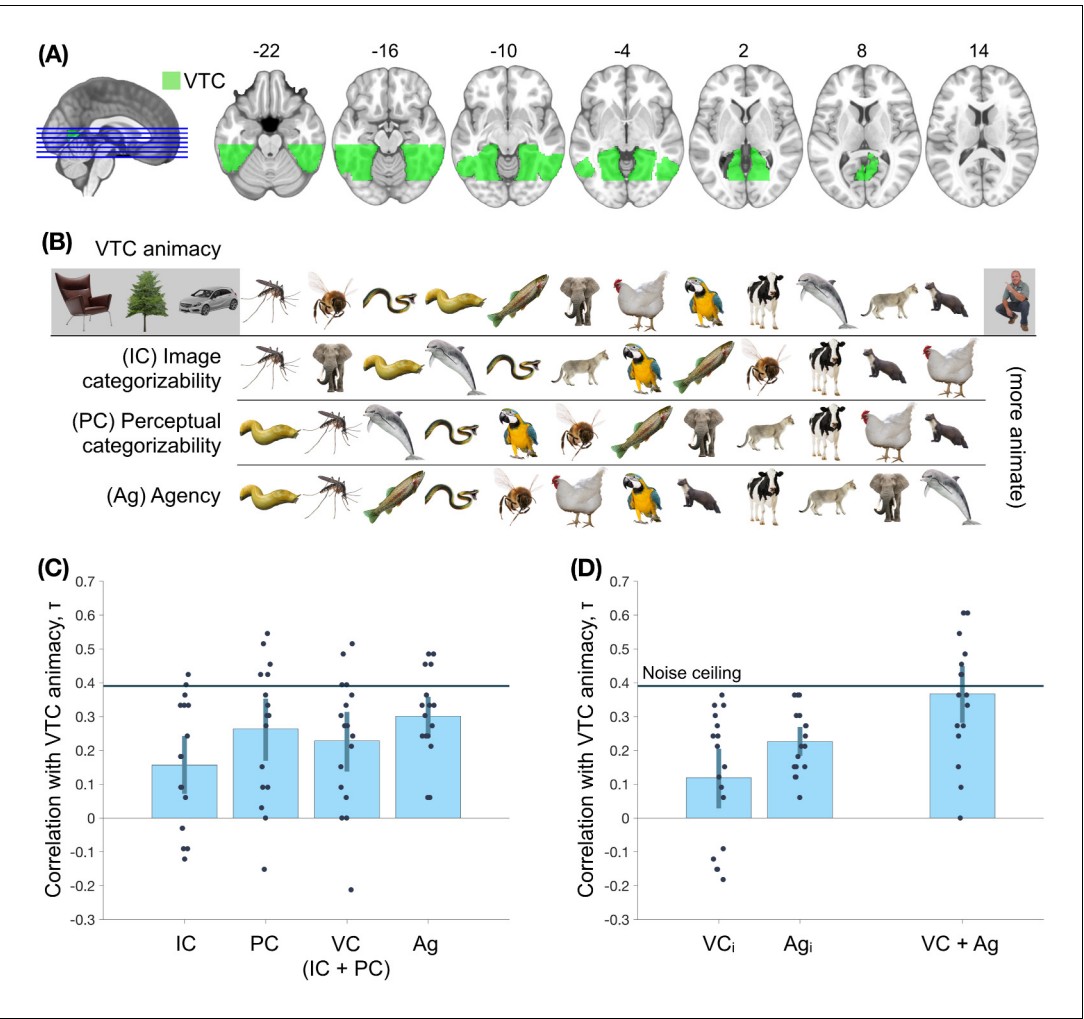

**Figure 4.** Assessing the nature of the animacy continuum in the ventral temporal cortex (VTC). (**A**) The region-of-interest, VTC, is highlighted. (**B**) The order of objects on the VTC animacy continuum, image categorizability (IC), perceptual categorizability (PC), and agency (Ag) are shown. (**C**) The within-participant correlations between VTC animacy and image categorizability (IC), perceptual categorizability (PC), visual categorizability (VC, a combination of image categorizability and perceptual categorizability; Materials and methods), and agency (Ag) are shown. All four models correlated positively with VTC animacy. (**D**) The left panel shows the correlations between VTC animacy and VC and Ag after regressing out the other measure from VTC animacy. Both correlations are positive, providing evidence for independent contributions of both agency and visual categorizability. The right panel shows the correlation between VTC animacy and a combination of agency and visual categorizability (Materials and methods). The combined model does not differ significantly from the VTC animacy noise ceiling (Materials and methods). This suggests that visual categorizability and agency are sufficient to explain the animacy organization in VTC. Error bars indicate 95% confidence intervals for the mean correlations.

DOI: https://doi.org/10.7554/eLife.47142.007

The following source data and figure supplements are available for figure 4:

**Source data 1.** Values of the rank-order correlations shown in the figure, for each participant.
DOI: https://doi.org/10.7554/eLife.47142.011

**Figure supplement 1.** The contribution of the principal components of VTC activations to VTC animacy.
DOI: https://doi.org/10.7554/eLife.47142.008

**Figure supplement 2.** The contributions of image and perceptual categorizabilities (IC and PC), independent of agency (Ag), to VTC animacy.
DOI: https://doi.org/10.7554/eLife.47142.009

**Figure supplement 3.** The robustness of our findings to the choice of the layer of VGG-16 used to quantify image categorizability.

*Figure 4 continued on next page*

*Figure 4 continued*

DOI: https://doi.org/10.7554/eLife.47142.010

Because agency and visual categorizability scores were weakly correlated, it is possible that the contribution of one of these factors was partly driven by the other. To test for their independent contributions, we regressed out the contribution of the other factor(s) from VTC animacy scores before computing the correlations. The correlation between VTC animacy and agency remained positive in all individual participants ($\tau = 0.23$; $p < 10^{-4}$ *Figure 4D*) after regressing out both image categorizability and perceptual categorizability. Similarly, the correlation between VTC animacy and visual categorizability remained positive after regressing out agency ($\tau = 0.12$; $p = 4.7 \times 10^{3}$).

Finally, to test whether a combined model including visual categorizability and agency fully explained the animacy continuum, we performed leave-one-out regression on VTC animacy with all three factors as independent measures. The resultant combined model (derived separately for each left-out participant) had a higher correlation with VTC animacy than any of the three factors alone (within-participant comparisons - $\Delta_{IC} = 0.21$, $p < 10^{-4}$; $\Delta_{PC} = 0.10$, $p = 6 \times 10^{-4}$, $\Delta_{Ag} = 0.07$, $p = 8.3 \times 10^{-3}$). Furthermore, the correlation between the combined model and VTC animacy ($\tau = 0.37$; *Figure 4D*) is at VTC animacy noise ceiling ($\tau_{NC} = 0.39$; Materials and methods). This result suggests that a linear combination of the two models (visual categorizability and agency) fully explains the animacy continuum in VTC, but the single models alone do not.

## Whole-brain searchlight analysis

Our results indicate that both visual categorizability and agency contribute to the animacy continuum in VTC as a whole. Can these contributions be anatomically dissociated as well? To test this, we repeated the analyses in a whole-brain searchlight analysis (spheres of 100 proximal voxels). To reduce the number of comparisons, we constrained the analysis to clusters showing significant above-chance animacy classification (Materials and methods). Our aim was to reveal spheres showing independent contributions of visual categorizability or agency. To obtain the independent contribution of agency, we regressed out both image categorizability and perceptual categorizability from each sphere's animacy continuum and tested if the residue reflected agency. Similarly, to obtain the independent contribution of visual categorizability, we regressed out agency from the sphere's animacy continuum and tested if the residue reflected either image categorizability or perceptual categorizability. The resulting brain maps were corrected for multiple comparisons (Materials and methods).

Results (*Figure 5*) showed that both visual categorizability and agency explained unique variance in clusters of VTC, consistent with the region-of-interest analysis. Moreover, there was a consistent anatomical mapping of the two factors: the independent visual categorizability contribution (LH: 1584 mm$^3$, center Montreal Neurological Institute (MNI) coordinates: $x = -38$, $y = -80$, $z = 7$; RH: 7184 mm$^3$, center coordinates: $x = 41$, $y = -71$, $z = 1$) was located posterior to the independent agency contribution (LH: 592 mm$^3$, center coordinates: $x = -42$, $y = -56$, $z = -19$; RH: 4000 mm$^3$, center coordinates: $x = 39$, $y = -52$, $z = -12$), extending from VTC into the lateral occipital regions. The majority of the independent agency contribution was located in the anterior part of VTC. A similar posterior-anterior organization was observed in both hemispheres (*Figure 5B*), though stronger in the right hemisphere. These results provide converging evidence for independent contributions of visual categorizability and agency to the animacy continuum, and show that these factors explain the animacy continuum at different locations in the visual system.

## Discussion

The present study investigated the organizing principles underlying the animacy organization in human ventral temporal cortex. Our starting point was the observation that the animacy organization expresses as a continuum rather than a dichotomy (*Connolly et al., 2012*; *Sha et al., 2015*; *Carlson et al., 2014*), such that some animals evoke more animate-like response patterns than others. Our results replicate this continuum, with the most animate response patterns evoked by humans and mammals and the weakest animate response patterns evoked by insects and snakes

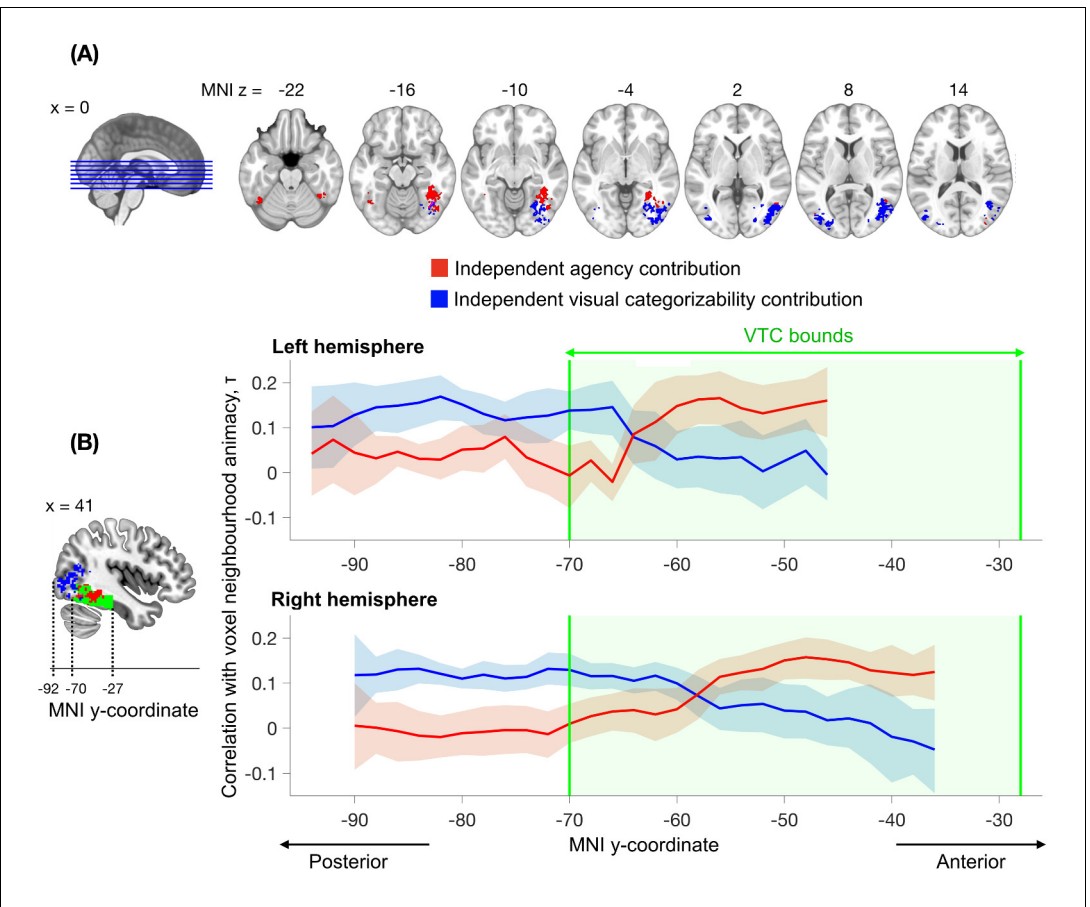

**Figure 5.** Searchlight analysis testing for the independent contributions of agency and visual categorizability to the animacy continuum. The analysis followed the approach performed within the VTC ROI (*Figure 4C*, middle panel) but now separately for individual spheres (100 voxels). The independent contribution of agency is observed within anterior VTC, while the independent contribution of visual categorizability extends from posterior VTC into the lateral occipital regions. Results are corrected for multiple comparisons (Materials and methods). (**B**) The correlations between agency and the animacy continuum in the searchlight spheres (variance independent of visual categorizability, in red) and the mean of the correlations between image and perceptual categorizabilities and the animacy continuum in the searchlight spheres (variance independent of agency, in blue), are shown as a function of the MNI y-coordinate. For each participant, the correlations are averaged across x and z dimensions for all the searchlight spheres that survived multiple comparison correction in the searchlight analysis depicted in (**A**). The blue and red bounds around the means reflect the 95% confidence bounds of the average correlations across participants. The green area denotes the anatomical bounds of VTC. Visual categorizability contributes more than agency to the animacy organization in the spheres in posterior VTC. This difference in contribution switches within VTC and agency contributes maximally to the animacy organization in more anterior regions of VTC.
DOI: https://doi.org/10.7554/eLife.47142.012

The following source data and figure supplement are available for figure 5:

**Source data 1.** Values of the correlations shown in the figure, for each participant, and the maps of the significant independent contributions of agency and visual categorizability across the brain.
DOI: https://doi.org/10.7554/eLife.47142.014

**Figure supplement 1.** The contributions (independent of agency) of image and perceptual categorizabilities to the animacy continuum in the searchlight spheres are shown (IC$_{Ag}$ and PC$_{Ag}$).
DOI: https://doi.org/10.7554/eLife.47142.013

(*Figure 4B*). Unlike previous studies, our stimulus set was designed to distinguish between two possible organizing principles underlying the animacy continuum, reflecting the degree to which an animal is visually animate (visual categorizability) and the degree to which an animal is perceived to have thoughts and feelings (agency). We found that both dimensions independently explained part

of the animacy continuum in VTC; together, they fully explained the animacy continuum. Whole-brain searchlight analysis revealed distinct clusters in which visual categorizability and agency explained the animacy continuum, with the agency-based organization located anterior to the visual categorizability-based organization. Below we discuss the implications of these results for our understanding of the animacy organization in VTC.

The independent contribution of visual categorizability shows that the animacy continuum in VTC is at least partly explained by the degree to which the visual features of an animal are typical of the animate category. This was observed both for the image features themselves (as represented in a CNN) and for the perceptual representations of these images in a behavioral task. These findings are in line with previous studies showing an influence of visual features on the categorical organization in high-level visual cortex (*Baldassi et al., 2013*; *Coggan et al., 2016*; *Nasr et al., 2014*; *Rice et al., 2014*; *Jozwik et al., 2016*). Furthermore, recent work has shown that mid-level perceptual features allow for distinguishing between animate and inanimate objects (*Levin et al., 2001*; *Long et al., 2017*; *Schmidt et al., 2017*; *Zachariou et al., 2018*) and that these features can elicit a VTC animacy organization in the absence of object recognition (*Long et al., 2018*). Our results show that (part of) the animacy continuum is similarly explained by visual features: animals that were more easily classified as animate by a CNN (based on visual features) were also more easily classified as animate in VTC. This correlation persisted when regressing out the perceived agency of the animals. Altogether, these findings support accounts that link the animacy organization in VTC to visual categorization demands (*Grill-Spector and Weiner, 2014*).

In parallel to investigations into the role of visual features in driving the categorical organization of VTC, other studies have shown that visual features do not full explain this organization (for reviews, see *Peelen and Downing, 2017*; *Bracci et al., 2017*). For example, animate-selective responses in VTC are also observed for shape- and texture-matched objects (*Proklova et al., 2016*; *Bracci and Op de Beeck, 2016*) and animate-like VTC responses can be evoked by geometric shapes that, through their movement, imply the presence of social agents (*Castelli et al., 2002*; *Martin and Weisberg, 2003*; *Gobbini et al., 2007*). The current results contribute to these findings by showing that (part of) the animacy continuum reflects the perceived agency of the animals: animals that were perceived as being relatively more capable of having thoughts and feelings were more easily classified as animate in VTC. This correlation persisted when regressing out the influence of animal-diagnostic visual features. These findings provide evidence that the animacy continuum is not fully explained by visual categorization demands, with perceived agency contributing significantly to the animacy organization. The finding of an agency contribution to the animacy continuum raises several interesting questions.

First, what do we mean with agency and how does it relate to other properties? In the current study, agency was measured as the perceived ability of an animal to think and feel. Ratings on these two scales were highly correlated with each other, and also likely correlate highly with related properties such as the ability to perform complex goal-directed actions, the degree of autonomy, and levels of consciousness (Appendix). On all of these dimensions, humans will score highest and animals that score highly will be perceived as relatively more similar to humans. As such, the agency contribution revealed in the current study may reflect a human-centric organization (*Contini et al., 2019*). Future studies could aim to disentangle these various properties.

Second, why would agency be an organizing principle? One reason for why agency could be an important organizing principle is because the level of agency determines how we interact with an animal: we can meaningfully interact with cats but not with slugs. To predict the behavior of high-agentic animals requires inferring internal states underlying complex goal-directed behavior (*Sha et al., 2015*). Again, these processes will be most important when interacting with humans but will also, to varying degrees, be recruited when interacting with animals. The agency organization may reflect the specialized perceptual analysis of facial and bodily signals that allow for inferring internal states, or the perceptual predictions that follow from this analysis.

Finally, how can such a seemingly high-level psychological property as agency affect responses in visual cortex? One possibility is that anterior parts of visual cortex are not exclusively visual and represent agency more abstractly, independent of input modality (*Fairhall et al., 2017*; *van den Hurk et al., 2017*). Alternatively, agency could modulate responses in visual cortex through feedback from downstream regions involved in social cognition. Regions in visual cortex responsive to social stimuli are functionally connected to the precuneus and medial prefrontal cortex – regions involved

in social perspective taking and reasoning about mental states (*Simmons and Martin, 2012*). Indeed, it is increasingly appreciated that category-selective responses in visual cortex are not solely driven by bottom-up visual input but are integral parts of large-scale domain-selective networks involved in domain-specific tasks like social cognition, tool use, navigation, or reading (*Peelen and Downing, 2017*; *Price and Devlin, 2011*; *Martin, 2007*; *Mahon and Caramazza, 2011*). Considering the close connections between regions within each of these networks, stimuli that strongly engage the broader system will also evoke responses in the visual cortex node of the network, even in the absence of visual input (*Peelen and Downing, 2017*; *Amedi et al., 2017*).

An alternative possibility is that agency is computed within the visual system based on visual features. This scenario is consistent with our results as long as these features are different from the features leading to animate-inanimate categorization. Similar to the visual categorizability of animacy, visual categorizability of agency could arise if there was strong pressure to quickly determine the agency of an animal based on visual input. In a supplementary analysis, we found that a model based on the features represented in the final fully connected layer of a CNN allows for predicting agency ratings (Appendix). As such, it remains possible that agency is itself visually determined.

The unique contributions of agency and visual categorizability were observed in different parts of VTC, with the agency cluster located anterior to the visual categorizability cluster. This posterior-anterior organization mirrors the well-known hierarchical organization of visual cortex. A similar posterior-anterior difference was observed in studies dissociating shape and category representations in VTC, with object shape represented posterior to object category (*Proklova et al., 2016*; *Bracci and Op de Beeck, 2016*). The finding that visual categorizability and agency expressed in different parts of VTC is consistent with multiple scenarios. One possibility is that VTC contains two distinct representations of animals, one representing category-diagnostic visual features and one representing perceived agency. Alternatively, VTC may contain a gradient from a more visual to a more conceptual representations of animacy, with the visual representation gradually being transformed into a conceptual representation. More work is needed to distinguish between these possibilities.

In sum, our results provide evidence that two principles independently contribute to the animacy organization in human VTC: visual categorizability, reflecting the presence of animal-diagnostic visual features, and agency, reflecting the degree to which animals are perceived as thinking and feeling social agents. These two principles expressed in different parts of visual cortex, following a posterior-to-anterior distinction. The finding of an agency organization that is not explained by differences in visual categorizability raises many new questions that can be addressed in future research.

## Materials and methods

### Neural network for image categorizability

Image categorizability was quantified using a convolutional neural network (CNN), VGG-16 (*Simonyan and Zisserman, 2015*), which had 13 convolutional layers and 3 fully-connected layers which map $224 \times 224 \times 3$ RGB images to a 1000 dimensional object category space (each neuron corresponds to distinct labels such as cat and car). The CNN was taken from the MatConvNet package (*Vedaldi and Lenc, 2015*) and was pre-trained on images from the ImageNet ILSVRC classification challenge (*Russakovsky et al., 2015*).

Activations were extracted from the final fully-connected layer, prior to the softmax operation, for 960 colored images obtained from *Kiani et al. (2007)* of which 480 contain an animal or animal parts and the rest contain inanimate objects or their parts. The dimensionality of the activations was reduced to 495 dimensions using principal component analysis in order to reduce the training time while keeping the captured variance high (more details can be found in *Thorat, 2017*). Support vector machines (SVMs) with linear kernels were trained (with the default parameters of the function fitcsvm in MATLAB r2017b, The MathWorks, Natick, MA) to distinguish between animate and inanimate object-driven activations. Training accuracies were quantified using 6-fold cross-validation. The average cross-validation accuracy was 92.2%. The image categorizability of an object was defined as the distance to the representation of that object from the decision boundary of the trained classifier. The stimuli used in the subsequent tasks, behavioral and fMRI, did not occur in this training set of 960 images.

## Visual search task for perceptual categorizability

Perceptual categorizability was quantified using a visual search task adopted from *Mohan and Arun (2012)*. 29 healthy adults participated in this experiment, of which 21 (9 females; age range: 19–62, median: 27) were selected according to the conditions specified below to obtain an estimate of perceptual categorizability. All participants gave informed consent. All procedures were carried out in accordance with the Declaration of Helsinki and were approved by the ethics committee of Radboud University (ECSW2017-2306-517).

For details of the experimental procedure, we refer the reader to *Proklova et al. (2016)*. Briefly, on each trial, participants were presented with 16 images and had to indicate, as fast and as accurately as possible, in which panel (left or right) the oddball image occurred (see *Figure 1C*). The 15 distractor images were identical except that they varied in size. Participants were not instructed to look for a particular category and only had to indicate the position of the different-looking object. All participants had accuracies of 90% and higher in every block. The trials on which they made an error were repeated at the end of the respective blocks. Psychtoolbox (*Brainard, 1997*) controlled the stimuli presentation. Gray-scaled images of the animate objects (four exemplars each of 12 animals and humans) used in the fMRI experiment and 72 images (36 animate) from the functional localizer experiment of *Proklova et al. (2016)* were used in this experiment. Images were gray-scaled to make the participants capitalize on differences in object shapes and textures rather than color.

In order to obtain perceptual categorizability scores for the animate objects used in the fMRI experiment, we trained animate-inanimate classifiers on representations capturing perceptual similarity between objects. For each participant, the images of animate objects from the fMRI experiment were the test set, and 28 (14 animate) images randomly chosen from the 72 images were the training set. In order to obtain representations which encoded perceptual similarity between objects, each of the images from the training and test set were used as either targets or distractors (randomly chosen for each participant) while the images from the training set were used as distractors or targets (corresponding to the previous choice made for each participant). The inverse of the reaction time was used as a measure of perceptual similarity (*Mohan and Arun, 2012*). For each of the images in the train and test set, 82 values (1/RT) were obtained which were associated with a perceptual similarity-driven representational space and were used as features for the animate-inanimate classifier. A linear SVM was trained to classify the training images as animate or inanimate. The distances of the representations of the test images were then calculated from the classification boundary and were termed decision scores. This resulted, for each participant, in decision scores for images of animals and humans used in the main fMRI experiment.

For further analysis, only those participants who had both training (4-fold cross-validation) and test accuracies for animacy classification above 50% were selected. For the relevant 21 participants, the mean training accuracy was 63.4% (>50%, $p < 10^{-4}$), and the mean test accuracy was 70.0% (>50%, $p < 10^{-4}$). Each object's perceptual categorizability was quantified as the average of its decision scores across participants.

## Main ratings experiment for agency

Agency was quantified using a ratings experiment. Sixteen healthy adults participated in this experiment (9 females; all students at the University of Trento). All participants gave informed consent. All procedures were carried out in accordance with the Declaration of Helsinki and were approved by the ethics committee of the University of Trento (protocol 2013–015).

In each trial, four colored images of an animal from a set of 40 animals were shown, and participants were asked to indicate, on a scale of 0 to 100, how much thoughtfulness or feelings they attributed to the animal, or how familiar they were with the animal. These three factors constituted three blocks of the experiment (the order was randomized across participants). At the beginning of each block, a description of the relevant factor was provided. Participants were encouraged to use the descriptions as guidelines for the three factors. In quantifying their familiarity with an animal, participants had to account for the knowledge about and the amount of interaction they have had with the animal. In quantifying the thoughtfulness an animal might have, participants had to account for the animal's ability in planning, having intentions, and abstraction. In quantifying the feelings an animal might have, participants had to account for the animal's ability to empathise, have sensations, and react to situations.

As mentioned in the Results section, the feelings and thoughtfulness ratings co-varied substantially with each other (the within-participant correlations were at the noise ceilings of the two factors). Agency was quantified as the average of the ratings for feelings and thoughtfulness.

## fMRI experiment

A functional magnetic resonance imaging (fMRI) experiment was performed to obtain the animacy continua in high-level visual cortex, specifically the ventral temporal cortex (VTC). The design was adopted from *Proklova et al. (2016)*. Schematics of the main experiment and the animacy localizer experiment are shown in *Figure 3*.

### Participants

Seventeen healthy adults (6 females; age range: $20 - 32$, median: 25) were scanned at the Center for Mind/Brain Sciences of the University of Trento. This sample size was chosen to be the same as in *Proklova et al. (2016)*, as our animacy localizer and main experiment procedure were similar to theirs. All participants gave informed consent. All procedures were carried out in accordance with the Declaration of Helsinki and were approved by the ethics committee of the University of Trento (protocol 2013–015).

### Main experiment procedure

The stimuli consisted of colored images (4 exemplars each) of the 12 animals for which image categorizability and agency were orthogonalized, humans, and three inanimate objects (cars, plants, and chairs). There were a total of 64 images.

The main experiment consisted of eight runs. Each run consisted of 80 trials that were composed of 64 object trials and 16 fixation-only trials. In object trials, a single stimulus was presented for 300 ms, followed by a 3700 ms fixation period. In each run, each of the 64 images appeared exactly once. In fixation-only trials, the fixation cross was shown for 4000 ms. Trial order was randomized, with the constraints that there were exactly eight one-back repetitions of the same category (e.g., two cows in direct succession) within the object trials and that there were no two fixation trials appearing in direct succession. Each run started and ended with a 16s fixation period, leading to a total run duration of 5.9 min. Participants were instructed to press a button whenever they detected a one-back object repetition.

### Animacy localizer experiment procedure

In addition to the main experiment, participants completed one run of a functional localizer experiment. During the localizer, participants viewed grey-scale images of 36 animate and 36 inanimate stimuli in a block design. Each block lasted 16s, containing 20 stimuli that were each presented for 400 ms, followed by a 400 ms blank interval. There were eight blocks of each stimulus category and four fixation-only blocks per run. The order of the first 10 blocks was randomized and then mirror-reversed for the other 10 blocks. Participants were asked to detect one-back image repetitions, which happened twice during every non-fixation block.

### fMRI acquisition

Imaging data were acquired using a MedSpec 4-T head scanner (Bruker Biospin GmbH, Rheinstetten, Germany), equipped with an eight-channel head coil. For functional imaging, T2*-weighted EPIs were collected (repetition time = 2.0s, echo-time = 33 ms, 73° flip-angle, 3 mm × 3 mm × 3 mm voxel size, 1 mm gap, 34 slices, 192 mm field of view, 64 × 64 matrix size). A high-resolution T1-weighted image (magnetization prepared rapid gradient echo; 1 mm × 1 mm × 1 mm voxel size) was obtained as an anatomical reference.

### fMRI data pre-processing

The fMRI data were analyzed using MATLAB and SPM8. During preprocessing, the functional volumes were realigned, co-registered to the structural image, re-sampled to a 2 mm × 2 mm × 2 mm grid, and spatially normalized to the Montreal Neurological Institute 305 template included in SPM8. No spatial smoothing was applied.

## Region of interest - Ventral temporal cortex

VTC was defined as in *Haxby et al. (2011)*. The region extended from −71 to −21 on the y-axis of the Montreal Neurological Institute (MNI) coordinates. The region was drawn to include the inferior temporal, fusiform, and lingual/parahippocampal gyri. The gyri were identified using Automated Anatomical Labelling (AAL) parcellation (*Tzourio-Mazoyer et al., 2002*).

## Obtaining the animacy continua from fMRI data

Animacy continua were extracted from parts of the brain (either a region of interest such as VTC or a searchlight sphere) with a cross-decoding approach. SVM classifiers were trained on the BOLD images obtained from the animate and inanimate blocks of the localizer experiment, and tested on the BOLD images obtained from the main experiment. The degree of animacy of an object is given by the distance of its representation from the classifier decision boundary. As the BOLD response is delayed by seconds after stimulus onset, we had to decide the latency of the BOLD images we wanted to base our analysis on. The classification test accuracy and the animacy continuum noise ceiling for the objects from the main experiment were higher for the BOLD images at 4s latency than both 6s latency and the average of the images at 4 and 6s latencies. Therefore, we based our analysis on the BOLD images at 4s latency. The findings remain unchanged across the latencies mentioned.

All the images of the brain shown in this article were rendered using MRIcron.

## Comparing models with the animacy continua in the brain

We compared the animacy continua in the brain (such as the animacy continuum in VTC and animacy continua in searchlight spheres) with image and perceptual categorizabilities (visual categorizability), agency, their combination, and their independent components. The comparisons were performed at participant-level with rank-order correlations (Kendall's τ). The comparison between an animacy continuum and the independent component of a model was performed by regressing out other models from the animacy continuum and correlating the residue with the model of interest, for each participant.

Given a participant, the comparison between an animacy continuum and the combination of models was computed as follows. The animacy continuum was modeled as a linear combination of the models (with linear regression) for the rest of the participants. The regression weights associated with each model in the combination across those participants were averaged, and the animacy continuum of the participant of interest was predicted using a linear combination of the models using the averaged weights. The predicted animacy continuum was then correlated with the actual animacy continuum of this participant. This procedure was implemented iteratively for each participant to get a group estimate of the correlation between an animacy continuum and a combination of models.

## Comparing visual categorizability with the animacy continua

The contribution of visual categorizability to an animacy continuum is gauged by the comparison between that animacy continuum and a combination of image and perceptual categorizabilities in a leave-one-participant-out analysis as mentioned above. The independent contribution of visual categorizability to an animacy continuum is gauged by regressing out agency from the image and perceptual categorizabilities and combining the residues to model the animacy continuum in a leave-one-participant-out analysis as mentioned above. When visual categorizability is to be regressed out of an animacy continuum (to obtain the independent contribution of agency), image and perceptual categorizabilities are regressed out. When visual categorizability is to be included in a combination of models, image and perceptual categorizabilities are added as models.

To assess if a model or a combination of models explained all the variance in the animacy continuum across participants, for each participant we tested if the correlation between the model or the animacy continuum predicted by the combined model (in a leave-one-out fashion as above) and the average of animacy continua of the other participants was lower than the correlation between the participant's animacy continuum and the average of animacy continua of the other participants. On the group level, if this one-sided test (see 'Statistical tests in use') was not significant ($p > 0.05$), we concluded that the correlation between the model or a combination of models hit the animacy

continuum noise ceiling and thus explained all the variance in the animacy continuum across participants. In the comparisons in *Figure 4C–D*, only the correlation between VTC animacy and the combination of visual categorizability and agency was at VTC animacy noise ceiling.

### Searchlight details

In the whole-brain searchlight analysis, the searchlight spheres contained 100 proximal voxels. SVM classifiers were trained to distinguish between the BOLD images, within the sphere, corresponding to animate and inanimate stimuli from the localizer experiment. The classifiers were tested on the BOLD images, within the sphere, from the main experiment. Threshold-free cluster enhancement (TFCE; *Smith and Nichols, 2009*) with a permutation test was used to correct for multiple comparisons of the accuracies relative to baseline (50%). Further analysis was constrained to the clusters which showed above-chance classification (between-subjects, $p < 0.05$, on both localizer and main experiment accuracies) of animate and inanimate objects. Within each searchlight sphere that survived the multiple comparisons correction, the animacy continuum was compared with image and perceptual categorizabilities (after regressing out agency) and agency (after regressing out both image and perceptual categorizabilities). Multiple comparisons across spheres of correlations to baseline (0) were corrected using TFCE. The independent visual categorizability clusters were computed as a union of spheres that had a significant contribution (independent of agency) from either image or perceptual categorizabilities.

### Statistical tests in use

Hypothesis testing was done with bootstrap analysis. We sampled 10,000 times with replacement from the observations being tested. p-Values correspond to one minus the proportion of sample means that are above or below the null hypothesis (corresponding to the test of interest). The p-values reported in the paper correspond to one-sided tests. The 95% confidence intervals were computed by identifying the values below and above which 2.5% of the values in the bootstrapped distribution lay. Exact p-values are reported except when means of all the bootstrap samples are higher or lower than hypothesized in which case we mention $p < 10^{-4}$.

## Acknowledgements

We thank Daniel Kaiser for his help with experimental design. The research was supported by the Autonomous Province of Trento, Call 'Grandi Progetti 2012', project 'Characterizing and improving brain mechanisms of attention - ATTEND'. This project has received funding from the European Research Council (ERC) under the European Union's Horizon 2020 research and innovation programme (grant agreement No. 725970).

## Additional information

### Funding

| Funder | Grant reference number | Author |
|---|---|---|
| Horizon 2020 Framework Programme | 725970 | Marius V Peelen |
| Autonomous Province of Trento | ATTEND | Marius V Peelen |

The funders had no role in study design, data collection and interpretation, or the decision to submit the work for publication.

### Author contributions

Sushrut Thorat, Conceptualization, Investigation, Methodology, Writing—original draft, Writing—review and editing; Daria Proklova, Conceptualization, Resources; Marius V Peelen, Conceptualization, Supervision, Funding acquisition, Writing—original draft, Writing—review and editing

## Author ORCIDs

Marius V Peelen (iD) https://orcid.org/0000-0002-4026-7303

## Ethics

Human subjects: All participants gave informed consent. All procedures were carried out in accordance with the Declaration of Helsinki and were approved by the ethics committees of the University of Trento (protocol 2013-015) and the Radboud University (ECSW2017-2306-517).

## Decision letter and Author response

Decision letter https://doi.org/10.7554/eLife.47142.020
Author response https://doi.org/10.7554/eLife.47142.021

# Additional files

## Supplementary files

• Transparent reporting form
DOI: https://doi.org/10.7554/eLife.47142.015

## Data availability

All data and analysis code needed to reproduce the results reported in the manuscript can be found on OSF: https://doi.org/10.17605/OSF.IO/VXWG9. Source data files have been provided for Figures 2, 4 and 5.

The following dataset was generated:

| Author(s) | Year | Dataset title | Dataset URL | Database and Identifier |
|---|---|---|---|---|
| Thorat S, Peelen M | 2019 | The nature of the animacy organization in human ventral temporal cortex - Essential data and analysis code | https://doi.org/10.17605/OSF.IO/VXWG9 | Open Science Framework, 10.17605/OSF.IO/VXWG9 |

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

## Appendix 1

DOI: https://doi.org/10.7554/eLife.47142.016

### Agency can be derived from visual feature differences

To test whether agency ratings can be predicted based on high-level visual feature representations, agency ratings were collected for a set of 436 images. The activation patterns of these images in the final fully-connected layer (FC8) of VGG-16 was established. A regression model trained on these activation patterns could accurately predict agency ratings of the stimuli used in our fMRI experiment, as described in more detail below.

Agency ratings were collected for 436 object images, which included the 12 animal images from the main fMRI experiment. The ratings experiment was similar to the main ratings experiment. However, instead of thoughtfulness and feelings, 16 participants rated the agency of animals shown in the stimuli. All participants gave informed consent. All procedures were carried out in accordance with the Declaration of Helsinki and were approved by the ethics committee of Radboud University (ECSW2017-2306-517). One image of an object was shown at a time. Agency was defined as the capacity of individuals to act independently and to make their own free choices, and participants were instructed to consider factors such as 'planning, intentions, abstraction, empathy, sensation, reactions, thoughtfulness, feelings'. The agency ratings for the 12 animals co-varied positively with the agency scores from the main ratings experiment ($\tau = 0.75$, $p<10^{-4}$), and the mean correlation was at (main ratings experiment's) agency noise ceiling ($\tau = 0.75$).

Activations from VGG-16 FC8 were extracted for these 436 images and principal component analysis was performed on the activations driven by the 388 images, excluding the $12 \times 4$ animal images from the main fMRI experiment. A cross-validated regression analysis was performed, with the agency ratings as the dependent variable and the principal components of FC8 as the independent variables. The first 20 principal components (regularisation cut-off) were included in the final model, as the models with more components provided with little gains in the similarities of the computed scores to the actual agency scores for the left-out images, while the similarities for the included images kept increasing (over-fitting). The agency scores were computed for the left out 12 animals and compared to the agency ratings obtained from the current experiment. They co-varied positively ($\tau = 0.61$, $p<10^{-4}$) but the mean correlation was not at the agency ratings noise ceiling ($\tau = 0.79$). These observations show that agency ratings can be predicted based on high-level visual feature representations in a feedforward convolutional neural network.

