## [Decision Letter]

Thank you for submitting your article "The nature of the animacy organization in human ventral temporal cortex" for consideration by *eLife*. Your article has been reviewed by three peer reviewers, including Thomas Serre as the Reviewing Editor and Reviewer #1, and the evaluation has been overseen by Michael Frank as the Senior Editor. The following individuals involved in the review of your submission have agreed to reveal their identity: James Haxby (Reviewer #2); Dirk Bernhardt-Walther (Reviewer #3).

The reviewers have discussed the reviews with one another and the Reviewing Editor has drafted this decision to help you prepare a revised submission.

Summary:

This is a timely manuscript that seeks to explain the animacy continuum found in the Ventral Temporal Cortex (VTC). The authors used computational and behavioral methods to parameterize a set of stimuli used for an fMRI experiment to disentangle visual categorizability vs. agency. While these two measures have been confounded in previous work, the present study uses a stimulus set in which visual categorizability and agency are dissociated based on representations derived from convolutional neural networks and behavioral experiments. A linear model which incorporates both measures fully explains the continuum found in the fMRI experiment. A subsequent spotlight search shows that categorizability and agency are actually represented in separate clusters of voxels further strengthening the main result.

The reviewers unanimously agreed that the paper is of sufficient interest, quality, and novelty to merit accepting it for publication in *eLife*. The reviewers all agreed that no additional experiments are needed. A revision is however needed to clarify some of the methods and concepts used as well as to provide additional data analyses. We have the following suggestions to improve the manuscript.

Essential revisions:

The reviewers suggest further analyses and figures to clarify how the representation of visual categorization and of agency are structured. Is the animacy continuum based both on visual features and agentic properties, or are there two animacy continua or a gradient reflecting a transition from a representation of visual information that lays the foundation and a representation of agency? Should the term "animacy continuum" refer to an amalgam of visual features and agency, or should it be reserved for semantic content?

One suggestion includes a multidimensional scaling analysis similar to the analyses and figures of Connolly et al. (2016). In the original paper, the authors showed gradients of changes in representational geometry that (a) disentangled representation of animal taxonomy vs. visual appearance in VTC and (b) disentangled predacity from visual appearance and taxonomy along the STS. A similar method could be used to help clarify exactly how the animacy continuum in VTC reflects both visual categorizability and agency. They seem to be dissociable, and the anatomical and conceptual aspects of this dissociation could be explored and explicated more thoroughly.

In a similar vein, the authors pose the question "How can such a seemingly high-level psychological property as agency affect responses in visual cortex?" Why the need to assume VTC is purely visual? This assumption carries with it the assertion that semantic information must come from elsewhere in a large-scale network. It seems quite possible that IT cortex makes the same computations to extract this critical feature from the constellation of visual features and learned associations that could be calculated elsewhere in a network, and direct and automatic extraction of this information in VTC may be more efficient and adaptive, as the authors acknowledge later in the Discussion. Are such computations "visual" or should VTC be characterized as something more than "visual"? The authors cite numerous papers that show the influence of nonvisual information on representation in VTC, and clearer incorporation of these facts in their reasoning could help here. Incidentally, the discussion of nonvisual activation of VTC should include a paper by Fairhall et al. (2017), who show representation in VTC of agentic properties conveyed by voices in the congenitally blind.

With regard to using a CNN as a measure of visual categorizablity, the authors write, "Because this measure was based on a feedforward transformation of the images, we label this measure image categorizability." This statement may not be completely accurate. The training of a CNN uses millions of semantically-labeled images, meaning that semantics are incorporated in the development of feature spaces that afford view-invariant, lighting-invariant, and most importantly exemplar-invariant labeling of new images. Thus, characterizing this CNN as solely feedforward overlooks the fact that it is trained with semantic labels and produces semantic labels. This gray area between "visual categorizability" and semantic classification needs to be clarified. The current manuscript tries to make a binary distinction between visual categorizability and semantic judgment of agency, when in fact it is more complicated insofar as both the CNN and behavioral criteria for visual categorizability can be influenced by semantics – the CNN in how it is trained and the behavioral task in terms of the role that automatic activation of semantic features and categories may influence response times.

The reviewers suggest a rework of the Materials and methods section as they found it difficult to decipher what exactly had been done in the experiment and in the data analysis. Below is a list of clarifications or requests for missing information.

Clarifications regarding methods for using the CNN are also requested:

1) Please confirm/clarify that the network was not specifically fine-tuned for animal/non-animal categorization. An SVM is trained for animal vs. non-animal categorization on top of features extracted from a pre-trained CNN.

2) More generally, when using a CNN to get a visual categorizability score per image, the choice of the final layer as "feature representation" is somewhat odd because units are already category selective and responses across category units tend to be very sparse. For these kinds of transfer learning tasks, people normally consider layers below. Prior work (see Eberhardt et al., 2016) has shown that the best correlation between layer outputs and human decisions for rapid animal/non-animal categorization was actually in the higher convolutional layers. Please comment.

3) In the statement: "Training accuracies were quantified using 6-fold cross-validation. The training accuracy was 92.2%." The authors probably meant "average test or validation accuracy", right (i.e., by training using 5 folds and testing on the remaining one and averaging over 6 rounds)? Please confirm.

4) The image categorizability of an object was defined as the distance to the representation of that object from the decision boundary of the trained classifier. Please confirm that this is distance when the image is used as a test.

5) There is a methodological difference in the way categorizability scores are derived for CNNs and humans. Why? It would have been better to use the cross-validation method for both but obviously, there are constraints on how perceptual measures can be derived. Why not then use the same method for CNNs as for the human experiment of categorizability?

Clarifications regarding human experiments:

1) Which classifier was used for the animacy decoding?

2) What is the "animacy continuum"? Is it the unthresholded decision value of the classifier that was trained on the localizer run?

3) The authors introduced two separate measures of categorizability, which turn out to be correlated. How do these two separate measures of categorizability interact with animacy?

4) The linear relationship between animacy, agency, and categorizability could be investigated solely based on behavioral data. What was the scientific contribution that we obtained from the fMRI data? There is no question that there is one, but we would like to see this worked out better.

5) There is no statement about informed consent for the psychophysics experiments.

6) The image used differ across experiments: sometimes images are grayscale, sometimes color and sometimes unspecified. Please comment.

7) The subsection about "Comparing visual categorizability with the animacy continua" is confusing. As written, the statistical analysis tests whether the model correlation with the animacy score is *lower* than that of the human to human correlation. If the corresponding p-value is <.05 as stated, then the model is indeed lower than the noise ceiling not at the noise ceiling? Please clarify. At the very least the wording should be such as to remove any source of ambiguity about the null hypothesis etc.

8) It is said that "Animacy classifiers were trained on the BOLD images obtained from the animate and inanimate blocks of the localizer experiment and tested on the BOLD images obtained from the main experiment." How do you then get a p-value for significance for above chance classification for each sphere as reported in the subsection “Searchlight details”?

---

## [Author Response]

Essential revisions:The reviewers suggest further analyses and figures to clarify how the representation of visual categorization and of agency are structured. Is the animacy continuum based both on visual features and agentic properties, or are there two animacy continua or a gradient reflecting a transition from a representation of visual information that lays the foundation and a representation of agency? Should the term "animacy continuum" refer to an amalgam of visual features and agency, or should it be reserved for semantic content?

Thanks for raising these intriguing questions. In terms of terminology, we use “animacy continuum” to refer to the continuum observed in the representational distances from the decision boundary of the animate-inanimate classifier in VTC. As we (and others before us) show, not all animals are equally animate when considering SVM decision scores – some animals (e.g., mammals and humans) are very strongly animate while others (e.g., insects and reptiles) are weakly animate. As we outline in the Introduction of the paper, this continuum in visual cortex might be explained by differences in animate-diagnostic features, agency, or both. Our results show that the animacy continuum in VTC is explained by both visual features and agency. In a new analysis (Figure 5B), we confirm that these two factors contribute at different locations in the visual system. This is consistent with a gradient reflecting a transformation from visual features to agency. However, it is also consistent with the existence of two separate continua that appear as gradients due to overlap or cross-participant averaging. We have added these considerations to the Discussion: “The finding that visual categorizability and agency expressed in different parts of VTC is consistent with multiple scenarios. One possibility is that VTC contains two distinct representations of animals, one representing diagnostic visual features and one representing perceived agency. Alternatively, VTC may contain a gradient from a more visual to a more conceptual representations of animacy, with the visual representation gradually being transformed into a conceptual representation. More work is needed to distinguish between these possibilities.”

One suggestion includes a multidimensional scaling analysis similar to the analyses and figures of Connolly et al. (2016). In the original paper, the authors showed gradients of changes in representational geometry that (a) disentangled representation of animal taxonomy vs. visual appearance in VTC and (b) disentangled predacity from visual appearance and taxonomy along the STS. A similar method could be used to help clarify exactly how the animacy continuum in VTC reflects both visual categorizability and agency. They seem to be dissociable, and the anatomical and conceptual aspects of this dissociation could be explored and explicated more thoroughly.

We have added new analyses in response to these suggestions:

1) Figure 5B shows the contributions of visual categorizability and agency to the animacy continuum along the posterior-anterior axis. Confirming the searchlight maps, we observed a clear posterior-to-anterior transition.

2) We have added a principal component analysis on VTC patterns, similar to Connolly et al. (2016) to Figure 4 (Figure 4—figure supplement 1). This analysis, while not directly testing our hypothesis, visualizes the main dimensions of VTC representations more generally, as well as how these relate to VTC animacy.

In a similar vein, the authors pose the question "How can such a seemingly high-level psychological property as agency affect responses in visual cortex?" Why the need to assume VTC is purely visual? This assumption carries with it the assertion that semantic information must come from elsewhere in a large-scale network. It seems quite possible that IT cortex makes the same computations to extract this critical feature from the constellation of visual features and learned associations that could be calculated elsewhere in a network, and direct and automatic extraction of this information in VTC may be more efficient and adaptive, as the authors acknowledge later in the Discussion. Are such computations "visual" or should VTC be characterized as something more than "visual"? The authors cite numerous papers that show the influence of nonvisual information on representation in VTC, and clearer incorporation of these facts in their reasoning could help here. Incidentally, the discussion of nonvisual activation of VTC should include a paper by Fairhall et al. (2017), who show representation in VTC of agentic properties conveyed by voices in the congenitally blind.

We agree that it is possible that VTC itself is not exclusively visual. We now added this possibility to the cited paragraph, which now also includes a citation to the Fairhall et al. paper: “One possibility is that anterior parts of visual cortex are not exclusively visual and represent agency more abstractly, independent of input modality (Fairhall et al., 2017; van den Hurk et al., 2017). Alternatively, agency could modulate responses in visual cortex through feedback from downstream regions involved in social cognition.”

With regard to using a CNN as a measure of visual categorizablity, the authors write, "Because this measure was based on a feedforward transformation of the images, we label this measure image categorizability." This statement may not be completely accurate. The training of a CNN uses millions of semantically-labeled images, meaning that semantics are incorporated in the development of feature spaces that afford view-invariant, lighting-invariant, and most importantly exemplar-invariant labeling of new images. Thus, characterizing this CNN as solely feedforward overlooks the fact that it is trained with semantic labels and produces semantic labels. This gray area between "visual categorizability" and semantic classification needs to be clarified. The current manuscript tries to make a binary distinction between visual categorizability and semantic judgment of agency, when in fact it is more complicated insofar as both the CNN and behavioral criteria for visual categorizability can be influenced by semantics – the CNN in how it is trained and the behavioral task in terms of the role that automatic activation of semantic features and categories may influence response times.

We agree that it is likely that there are semantic influences on our visual measures, if only because the CNN was trained with semantically-labeled images. We have changed the mentioned statement to: "Because this measure was based on a feedforward transformation of the images which was not informed by inferred agentic properties of the objects (such as thoughtfulness), we label this measure image categorizability.” Importantly, in our work, we are interested in a specific part of semantic information – agency. VGG-16 is trained to classify images but is not provided with any kind of information about the agency of the objects in the images.

The behavioral task might also show some influence of semantic information. We now mention this in the text: “The neural representations the reaction times in this task rely on are not fully known, and might reflect information about inferred agency of the objects. As such, accounting for the contribution of perceptual categorizability in subsequent analyses provides a conservative estimate of the independent contribution of agency to neural representations in VTC.”

The reviewers suggest a rework of the Materials and methods section as they found it difficult to decipher what exactly had been done in the experiment and in the data analysis. Below is a list of clarifications or requests for missing information.Clarifications regarding methods for using the CNN are also requested:1) Please confirm/clarify that the network was not specifically fine-tuned for animal/non-animal categorization. An SVM is trained for animal vs. non-animal categorization on top of features extracted from a pre-trained CNN.

Yes, we trained an SVM on top of features of a pre-trained CNN. This is mentioned as follows: “The CNN was taken from the MatConvNet package (Vedaldi and Lenc, 2015) and was pre-trained on images from the ImageNet ILSVRC classification challenge (Russakovsky et al., 2015). Activations were extracted from the final fully-connected layer, prior to the softmax operation, for 960 colored images obtained from Kiani et al. (2007) of which 480 contain an animal or animal parts and the rest contain inanimate objects or their parts.”

2) More generally, when using a CNN to get a visual categorizability score per image, the choice of the final layer as "feature representation" is somewhat odd because units are already category selective and responses across category units tend to be very sparse. For these kinds of transfer learning tasks, people normally consider layers below. Prior work (see Eberhardt et al., 2016) has shown that the best correlation between layer outputs and human decisions for rapid animal/non-animal categorization was actually in the higher convolutional layers. Please comment.

We chose to focus on FC8 based on our previous (unpublished) work in which we observed that the neural representations in VTC correlated as highly as any other layer with the neural representations in FC8 of VGG-16, and that the animal/non-animal classification performance was one of the highest in FC8.

Following the reviewer’s suggestion, and motivated by Eberhardt et al., we ran the main analysis again using C5-2 features to compute image categorizability. Results are reported in Figure 4—figure supplement 3. As shown in the figure, results were highly similar when image categorizability (IC) was based on features of C5-2 rather than FC8. Specifically, the independent contributions of visual categorizability and agency to VTC animacy remained significant and the correlation between the combined model and VTC animacy was at VTC animacy noise ceiling.

We now also present the correlations between the image categorizability scores across all layers in Figure 1—figure supplement 1. This shows that the image categorizability in the fully connected layers (FC6-FC8) are highly similar. All the findings described in the paper are robust to a change in layer-selection among the fully connected layers.

3) In the statement: "Training accuracies were quantified using 6-fold cross-validation. The training accuracy was 92.2%." The authors probably meant "average test or validation accuracy", right (i.e., by training using 5 folds and testing on the remaining one and averaging over 6 rounds)? Please confirm.

Yes. We have changed the statement to: "The average cross-validation accuracy was 92.2%".

4) The image categorizability of an object was defined as the distance to the representation of that object from the decision boundary of the trained classifier. Please confirm that this is distance when the image is used as a test.

Yes. To clarify, we have added this sentence to the corresponding Materials and methods section: “The stimuli used in the subsequent tasks, behavioral and fMRI, did not occur in this training set of 960 images.”

5) There is a methodological difference in the way categorizability scores are derived for CNNs and humans. Why? It would have been better to use the cross-validation method for both but obviously, there are constraints on how perceptual measures can be derived. Why not then use the same method for CNNs as for the human experiment of categorizability?

In principle, the method used for extracting image categorizability scores from the CNN is the most straightforward. As noted by the reviewer, that method could not be used to extract perceptual categorizability scores. In computing perceptual categorizability we obtained a measure of visual similarity between two given images. There are numerous metrics which could be used to obtain such a measure of visual similarity between the representations of two images from a CNN (e.g. Pearson correlation, Euclidean distance, Isomap distance). Which of such metrics is appropriate is not a trivial consideration. Rather than choosing an arbitrary metric, we chose to stick to the most straightforward approach for the CNN. It should be noted that image and perceptual categorizability correlated quite strongly, despite the differences in the methodology of these measures.

Clarifications regarding human experiments1) Which classifier was used for the animacy decoding?

SVMs were used for animacy decoding, and this is now indicated in the sub-section “Obtaining the animacy continua from fMRI data”.

2) What is the "animacy continuum"? Is it the unthresholded decision value of the classifier that was trained on the localizer run?

Yes, as indicated by "The degree of animacy of an object is given by the distance of its representation from the classifier decision boundary." in the sub-section “Obtaining the animacy continua from fMRI data”.

3) The authors introduced two separate measures of categorizability, which turn out to be correlated. How do these two separate measures of categorizability interact with animacy?

We have added two new figures (Figure 4—figure supplement 2; Figure 5—figure supplement 1) to show the separate contributions of visual and perceptual categorizability to animacy after regressing out agency.

4) The linear relationship between animacy, agency, and categorizability could be investigated solely based on behavioral data. What was the scientific contribution that we obtained from the fMRI data? There is no question that there is one, but we would like to see this worked out better.

Our study aimed to improve our understanding of the organizing principles of human ventral temporal cortex. Without the fMRI data we would not have had access to the animacy continuum in VTC. Behavioral experiments can be used to obtain estimates of how animate an object is for humans, and how visual or conceptual features contribute to this, but would not tell us how those estimates relate to the animacy continuum in VTC.

5) There is no statement about informed consent for the psychophysics experiments.

We have added informed consent statements for every experiment mentioned.

6) The image used differ across experiments: sometimes images are grayscale, sometimes color and sometimes unspecified. Please comment.

We now mention whether the images were colored or gray-scaled in the methods description of each experiment.

7) The subsection about "Comparing visual categorizability with the animacy continua" is confusing. As written, the statistical analysis tests whether the model correlation with the animacy score is lower than that of the human to human correlation. If the corresponding p-value is <.05 as stated, then the model is indeed lower than the noise ceiling not at the noise ceiling? Please clarify. At the very least the wording should be such as to remove any source of ambiguity about the null hypothesis etc.

We have adjusted the wording: “To assess if a model or a combination of models explained all the variance in the animacy continuum across participants, for each participant we tested if the correlation between the model or the animacy continuum predicted by the combined model (in a leave-one-out one fashion as above) and the average of animacy continua of the other participants was lower than the correlation between the participant's animacy continuum and the average of animacy continua of the other participants. On the group level, if this one-sided test (see “Statistical tests in use") was not significant (*p* > 0.05), we concluded that the correlation between the model or a combination of models hit the animacy continuum noise ceiling and thus explained all the variance in the animacy continuum across participants.”

8) It is said that "Animacy classifiers were trained on the BOLD images obtained from the animate and inanimate blocks of the localizer experiment and tested on the BOLD images obtained from the main experiment." How do you then get a p-value for significance for above chance classification for each sphere as reported in the subsection “Searchlight details”?

The “Searchlight details” section has been updated to further clarify the methods. The p-value mentioned in the question is computed in a test of above-chance classification for each sphere *across participants*.